# Optimistic planning in Markov decision processes using a generative model

**Balázs Szörényi**
INRIA Lille - Nord Europe,
SequeL project, France /
MTA-SZTE Research Group on
Artificial Intelligence, Hungary
balazs.szorenyi@inria.fr

**Gunnar Kedenburg**
INRIA Lille - Nord Europe,
SequeL project, France
gunnar.kedenburg@inria.fr

**Remi Munos**[*]
INRIA Lille - Nord Europe,
SequeL project, France
remi.munos@inria.fr

## Abstract

We consider the problem of online planning in a Markov decision process with discounted rewards for any given initial state. We consider the PAC sample complexity problem of computing, with probability $1-\delta$, an $\epsilon$-optimal action using the smallest possible number of calls to the generative model (which provides reward and next-state samples). We design an algorithm, called StOP (for Stochastic-Optimistic Planning), based on the "optimism in the face of uncertainty" principle. StOP can be used in the general setting, requires only a generative model, and enjoys a complexity bound that only depends on the local structure of the MDP.

## 1 Introduction

### 1.1 Problem formulation

In a *Markov decision process* (MDP), an agent navigates in a state space $X$ by making decisions from some action set $U$. The dynamics of the system are determined by transition probabilities $P : X \times U \times X \to [0, 1]$ and reward probabilities $R : X \times U \times [0, 1] \to [0, 1]$, as follows: when the agent chooses action $u$ in state $x$, then, with probability $R(x, u, r)$, it receives reward $r$, and with probability $P(x, u, x')$ it makes a transition to a next state $x'$. This happens independently of all previous actions, states and rewards—that is, the system possesses the *Markov property*. See [20, 2] for a general introduction to MDPs. We do not assume that the transition or reward probabilities are fully known. Instead, we assume access to the MDP via a *generative model* (e.g. simulation software), which, for a state-action $(x, u)$, returns a reward sample $r \sim R(x, u, \cdot)$ and a next-state sample $x' \sim P(x, u, \cdot)$. We also assume the number of possible next-states to be bounded by $N \in \mathbb{N}$.

We would like to find an agent that implements a policy which maximizes the expected cumulative discounted reward $\mathbb{E}[\sum_{t=0}^{\infty} \gamma^t r_t]$, which we will also refer to as the *return*. Here, $r_t$ is the reward received at time $t$ and $\gamma \in (0, 1)$ is the *discount factor*. Further, we take an *online planning* approach, where at each time step, the agent uses the generative model to perform a simulated search (planning) in the set of policies, starting from the current state. As a result of this search, the agent takes a single action. An expensive global search for the optimal policy in the whole MDP is avoided.

---

[*]Current affiliation: Google DeepMind

To quantify the performance of our algorithm, we consider a PAC (Probably Approximately Correct) setting, where, given $\epsilon > 0$ and $\delta \in (0, 1)$, our algorithm returns, with probability $1 - \delta$, an $\epsilon$-optimal action (i.e. such that the loss of performing this action and then following an optimal policy instead of following an optimal policy from the beginning is at most $\epsilon$). The number of calls to the generative model required by the planning algorithm is referred to as its *sample complexity*. The sample and computational complexities of the planning algorithm introduced here depend on local properties of the MDP, such as the quantity of near-optimal policies starting from the initial state, rather than global features like the MDP's size.

## 1.2   Related work

The online planning approach and, in particular, its ability to get rid of the dependency on the global features of the MDP in the complexity bounds (mentioned above, and detailed further below) is the driving force behind the Monte Carlo Tree Search algorithms [16, 8, 11, 18]. [1] The theoretical analysis of this approach is still far from complete. Some of the earlier algorithms use strong assumptions, others are applicable only in restricted cases, or don't adapt to the complexity of the problem. In this paper we build on ideas used in previous works, and aim at fixing these issues.

A first related work is the sparse sampling algorithm of [14]. It builds a uniform look-ahead tree of a given depth (which depends on the precision $\epsilon$), using for each transition a finite number of samples obtained from a generative model. An estimate of the value function is then built using empirical averaging instead of expectations in the dynamic programming back-up scheme. This results in an algorithm with (problem-independent) sample complexity of order $\left(\frac{1}{(1-\gamma)^3 \epsilon}\right)^{\frac{\log K + \log[1/(\epsilon(1-\gamma)^2)]}{\log(1/\gamma)}}$ (neglecting some poly-logarithmic dependence), where $K$ is the number of actions. In terms of $\epsilon$, this bound scales as $\exp(O([\log(1/\epsilon)]^2))$, which is non-polynomial in $1/\epsilon$. [2] Another disadvantage of the algorithm is that the expansion of the look-ahead tree is uniform; it does not adapt to the MDP.

An algorithm which addresses this appears in [21]. It avoids evaluating some unnecessary branches of the look-ahead tree of the sparse sampling algorithm. However, the provided sample bound does not improve on the one in [14], and it is possible to show that the bound is tight (for both algorithms). In fact, the sample complexity turns out to be super-polynomial even in the pure Monte Carlo setting (i.e., when $K = 1$): $1/\epsilon^{2+(\log C)/\log(1/\gamma)}$, with $C \geq \frac{1}{\epsilon^2(1-\gamma)^4}$.

Close to our contribution are the planning algorithms [13, 3, 5, 15] (see also the survey [18]) that follow the so-called "optimism in the face of uncertainty" principle for online planning. This principle has been extensively investigated in the multi-armed bandit literature (see e.g. [17, 1, 4]). In the planning problem, this approach translates to prioritizing the most promising part of the policy space during exploration. In [13, 3, 5], the sample complexity depends on a measure of the quantity of near-optimal policies, which gives a better understanding of the real hardness of the problem than the uniform bound in [14].

The case of deterministic dynamics and rewards is considered in [13]. The proposed algorithm has sample complexity of order $(1/\epsilon)^{\frac{\log \kappa}{\log(1/\gamma)}}$, where $\kappa \in [1, K]$ measures (as a branching factor) the quantity of nodes of the planning tree that belong to near-optimal policies. If all policies are very good, many nodes need to be explored in order to distinguish the optimal policies from the rest, and therefore, $\kappa$ is close to the number of actions $K$, resulting in the minimax bound of $(1/\epsilon)^{\frac{\log K}{\log(1/\gamma)}}$. Now if there is structure in the rewards (e.g. when sub-optimal policies can be eliminated by observing the first rewards along the sequence), then the proportion of near-optimal policies is low, so $\kappa$ can be small and the bound is much better. In [3], the case of stochastic rewards have been considered. However, in that work the performance is not compared to the optimal (closed-loop) policy, but to the best open-loop policy (i.e. which does not depends on the state but only on the sequence of actions). In that situation, the sample complexity is of order $(1/\epsilon)^{\max\left(2, \frac{\log(\kappa)}{\log(1/\gamma)}\right)}$.

The deterministic and open-loop settings are relatively simple, since any policy can be identified with a sequence of actions. In the general MDP case however, a policy corresponds to an exponentially

wide tree, where several branches need to be explored. The closest work to ours in this respect is [5]. However, it makes the (strong) assumption that a full model of the rewards and transitions is available. The sample complexity achieved is again $\left(1/\epsilon\right)^{\frac{\log(\kappa)}{\log(1/\gamma)}}$, but where $\kappa \in (1, NK]$ is defined as the branching factor of the set of nodes that simultaneously (1) belong to near-optimal policies, and (2) whose "contribution" to the value function at the initial state is non-negligible.

## 1.3 The main results of the paper

Our main contribution is a planning algorithm, called StOP (for Stochastic Optimistic Planning) that achieves a polynomial sample complexity in terms of $\epsilon$ (which can be regarded as the leading parameter in this problem), and which is, in terms of this complexity, competitive to other algorithms that can exploit more specifics of their respective domains. It benefits from possible reward or transition probability structures, and does not require any special restriction or knowledge about the MDP besides having access to a generative model. The sample complexity bound is more involved than in previous works, but can be upper-bounded by:

$$(1/\epsilon)^{2 + \frac{\log \kappa}{\log(1/\gamma)} + o(1)} \tag{1}$$

The important quantity $\kappa \in [1, KN]$ plays the role of a branching factor of the set of important states $\mathcal{S}^{\epsilon,*}$ (defined precisely later) that "contribute" in a significant way to near-optimal policies. These states have a non-negligible probability to be reached when following some near-optimal policy. This measure is similar (but with some differences illustrated below) to the $\kappa$ introduced in the analysis of OP-MDP in [5]. Comparing the two, (1) contains an additional constant of $2$ in the exponent. This is a consequence of the fact that the rewards are random and that we do not have access to the true probabilities, only to a generative model generating transition and reward samples.

In order to provide intuition about the bound, let us consider several specific cases (the derivation of these bounds can be found in Section E):

- **Worst-case**. When there is no structure at all, then $\mathcal{S}^{\epsilon,*}$ may potentially be the set of all possible reachable nodes (up to some depth which depends on $\epsilon$), and its branching factor is $\kappa = KN$. The sample complexity is thus of order (neglecting logarithmic factors) $(1/\epsilon)^{2 + \frac{\log(KN)}{\log(1/\gamma)}}$. This is the same complexity that uniform planning algorithm would achieve. Indeed, uniform planning would build a tree of depth $h$ with branching factor $KN$ where from each state-action one would generate $m$ rewards and next-state samples. Then, dynamic programming would be used with the empirical Bellman operator built from the samples. Using Chernoff-Hoeffding bound, the estimation error is of the order (neglecting logarithms and $(1-\gamma)$ dependence) of $1/\sqrt{m}$. So for a desired error $\epsilon$ we need to choose $h$ of order $\log(1/\epsilon)/\log(1/\gamma)$, and $m$ of order $1/\epsilon^2$ leading to a sample complexity of order $m(KN)^h = (1/\epsilon)^{2 + \frac{\log(KN)}{\log(1/\gamma)}}$. (See also [15]) Note that in the worst-case sense there is no uniformly better strategy than a uniform planning, which is achieved by StOP. However, StOP can also do much better in specific settings, as illustrated next.

- **Case with $K_0 > 1$ actions at the initial state, $K_1 = 1$ actions for all other states, and arbitrary transition probabilities**. Now each branch corresponds to a single policy. In that case one has $\kappa = 1$ (even though $N > 1$) and the sample complexity of StOP is of order $\tilde{O}(\log(1/\delta)/\epsilon^2)$ with high probability[3]. This is the same rate as a Monte-Carlo evaluation strategy would achieve, by sampling $O(\log(1/\delta)/\epsilon^2)$ random trajectories of length $\log(1/\epsilon)/\log(1/\gamma)$. Notice that this result is surprisingly different from OP-MDP which has a complexity of order $(1/\epsilon)^{\frac{\log N}{\log(1/\gamma)}}$ (in the case when $\kappa = N$, i.e., when all transitions are uniform). Indeed, in the case of uniform transition probabilities, OP-MDP would sample the nodes in breadth-first search way, thus achieving this minimax-optimal complexity.

  This does not contradict the $\tilde{O}(\log(1/\delta)/\epsilon^2)$ bound for StOP (and Monte-Carlo) since this bound applies to an individual problem and holds in high probability, whereas the bound for OP-MDP is deterministic and holds uniformly over all problems of this type.

Here we see the potential benefit of using StOP instead of OP-MDP, even though StOP only uses a generative model of the MDP whereas OP-MDP requires a full model.

- **Highly structured policies**. This situation holds when there is a substantial gap between near optimal policies and other sub-optimal policies. For example if along an optimal policy, all immediate rewards are 1, whereas as soon as one deviates from it, all rewards are $< 1$. Then only a small proportion of the nodes (the ones that contribute to near-optimal policies) will be expanded by the algorithm. In such cases, $\kappa$ is very close to 1 and in the limit, we recover the previous case when $K = 1$ and the sample complexity is $O(1/\epsilon)^2$.

- **Deterministic MDPs**. Here $N = 1$ and we have that $\kappa \in [1, K]$. When there is structure in the rewards (like in the previous case), then $\kappa = 1$ and we obtain a rate $\tilde{O}(1/\epsilon^2)$. Now when the MDP is almost deterministic, in the sense that $N > 1$ but from any state-action, there is one next-state probability which is close to 1, then we have almost the same complexity as in the deterministic case (since the nodes that have a small probability to be reached will not contribute to the set of important nodes $\mathcal{S}^{\epsilon,*}$, which characterizes $\kappa$).

- **Multi-armed bandit** we essentially recover the result of the Action Elimination algorithm [9] for the PAC setting.

Thus we see that in the worst case StOP is minimax-optimal, and in addition, StOP is able to benefit from situations when there is some structure either in the rewards or in the transition probabilities. We stress that StOP achieves the above mentioned results *having no knowledge about $\kappa$*.

### 1.4 The structure of the paper

Section 2 describes the algorithm, and introduces all the necessary notions. Section 3 presents the consistency and sample complexity results. Section 4 discusses run time efficiency, and in Section 5 we make some concluding remarks. Finally, the supplementary material provides the missing proofs, the analysis of the special cases, and the necessary fixes for the issues with the run-time complexity.

## 2 `StOP`: Stochastic Optimistic Planning

Recall that $N \in \mathbb{N}$ denotes the number of possible next states. That is, for each state $x \in X$ and each action $u$ available at $x$, it holds that $P(x, u, x') = 0$ for all but at most $N$ states $x' \in X$. Throughout this section, the state of interest is denoted by $x_0$, the requested accuracy by $\epsilon$, and the confidence parameter by $\delta_0$. That is, the problem to be solved is to output an action $u$ which is, with probability at least $(1 - \delta_0)$, at least $\epsilon$-optimal in $x_0$.

The algorithm and the analysis make use of the notion of an (infinite) planning tree, policies and trajectories. These notions are introduced in the next subsection.

### 2.1 Planning trees and trajectories

The *infinite planning tree* $\mathbf{\Pi}^\infty$ for a given MDP is a rooted and labeled infinite tree. Its root is denoted $s_0$ and is labeled by the state of interest, $x_0 \in X$. Nodes on even levels are called *action nodes* (the root is an action node), and have $K_d$ children each on the $d$-th level of action nodes: each action $u$ is represented by exactly one child, labeled $u$. Nodes on odd levels are called *transition nodes* and have $N$ children each: if the label of the parent (action) node is $x$, and the label of the transition node itself is $u$, then for each $x' \in X$ with $P(x, u, x') > 0$ there is a corresponding child, labeled $x'$. There may be children with probability zero, but no duplicates.

An *infinite policy* is a subtree of $\mathbf{\Pi}^\infty$ with the same root, where each action node has exactly one child and each transition node has $N$ children. It corresponds to an agent having fixed all its possible future actions. A *(partial) policy* $\Pi$ is a finite subtree of $\mathbf{\Pi}^\infty$, again with the same root, but where the action nodes have *at most* one child, each transition node has $N$ children, and all leaves [4] are on the same level. The number of transition nodes on any path from the root to a leaf is denoted $d(\Pi)$ and is called the *depth* of $\Pi$. A partial policy corresponds to the agent having its possible future actions planned for $d(\Pi)$ steps. There is a natural partial order over these policies: a policy

$\Pi'$ is called *descendant policy* of a policy $\Pi$ if $\Pi$ is a subtree of $\Pi'$. If, additionally, it holds that $d(\Pi') = d(\Pi) + 1$, then $\Pi$ is called the *parent policy* of $\Pi'$, and $\Pi'$ the *child policy* of $\Pi$.

A *(random) trajectory*, or *rollout*, for some policy $\Pi$ is a realization $\tau := (x_t, u_t, r_t)_{t=0}^{T}$ of the stochastic process that belongs to the policy. A random path is generated from the root by always following, from a non-leaf action node with label $x_t$, its unique child in $\Pi$, then setting $u_t$ to the label of this node, from where, drawing first a label $x_{t+1}$ from $P(x_t, u_t, \cdot)$, one follows the child with label $x_{t+1}$. The reward $r_t$ is drawn from the distribution determined by $R(x_t, u_t, \cdot)$. The *value of the rollout* $\tau$ (also called return or payoff in the literature) is $v(\tau) := \sum_{t=0}^{T} r_t \gamma^t$, and the *value of the policy* $\Pi$ is $v(\Pi) := \mathbb{E}[v(\tau)] = \mathbb{E}[\sum_{t=0}^{T} r_t \gamma^t]$. For an action $u$ available at $x_0$, denote by $v(u)$ the maximum of the values of the policies having $u$ as the label of the child of root $s_0$. Denote by $v^*$ the maximum of these $v(u)$ values. Using this notation, the task of the algorithm is to return, with high probability, an action $u$ with $v(u) \geq v^* - \epsilon$.

## 2.2 The algorithm

StOP (Algorithm 1, see Figure 1 in the supplementary material for an illustration) maintains for each action $u$ available at $x_0$ a set of *active policies* $\text{Active}(u)$. Initially, it holds that $\text{Active}(u) = \{\Pi_u\}$, where $\Pi_u$ is the shallowest partial policy with the child of the root being labeled $u$. Also, for each policy $\Pi$ that becomes a member of an active set, the algorithm maintains high confidence lower and upper bounds for the value $v(\Pi)$ of the policy, denoted $\nu(\Pi)$ and $b(\Pi)$, respectively.

In each round $t$, an *optimistic policy* $\Pi_{t,u}^{\dagger} := \text{argmax}_{\Pi \in \text{Active}(u)} b(\Pi)$ is determined for each action $u$. Based on this, the current *optimistic action* $u_t^{\dagger} := \text{argmax}_u b(\Pi_{t,u}^{\dagger})$ and *secondary action* $u_t^{\dagger\dagger} := \text{argmax}_{u \neq u_t^{\dagger}} b(\Pi_{t,u}^{\dagger})$ are computed. A policy $\Pi_t$ to explore is then chosen: if the policy that belongs to the secondary action is at least as deeply developed as the policy that belongs to the optimistic action, the optimistic one is chosen for exploration, otherwise the secondary one. Note that a smaller depth is equivalent to a larger gap between lower and upper bound, and vice versa[5]. The set $\text{Active}(u_t)$ is then updated by replacing the policy $\Pi_t$ by its child policies. Accordingly, the upper and lower bounds for these policies are computed. The algorithm terminates when $\nu(\Pi_t^{\dagger}) + \epsilon \geq \max_{u \neq u_t^{\dagger}} b(\Pi_{t,u}^{\dagger})$–that is, when, with high confidence, no policies starting with an action different from $u_t^{\dagger}$ have the potential to have significantly higher value.

### 2.2.1 Number and length of trajectories needed for one partial policy

Fix some integer $d > 0$ and let $\Pi$ be a partial policy of depth $d$. Let, furthermore, $\Pi'$ be an infinite policy that is a descendant of $\Pi$. Note that

$$0 \leq v(\Pi') - v(\Pi) \leq \frac{\gamma^d}{1-\gamma}. \tag{2}$$

The value of $\Pi$ is a $\frac{\gamma^d}{1-\gamma}$-accurate approximation of the value of $\Pi'$. On the other hand, having $m$ trajectories for $\Pi$, their average reward $\hat{v}(\Pi)$ can be used as an estimate of the value $v(\Pi)$ of $\Pi$. From the Hoeffding bound, this estimate has, with probability at least $(1 - \delta)$, accuracy $\frac{1-\gamma^d}{1-\gamma} \sqrt{\frac{\ln(1/\delta)}{2m}}$.

With $m := m(d, \delta) := \lceil \frac{\ln(1/\delta)}{2} (\frac{1-\gamma^d}{\gamma^d})^2 \rceil$ trajectories, $\frac{\gamma^d}{1-\gamma} \geq \frac{1-\gamma^d}{1-\gamma} \sqrt{\frac{\ln(1/\delta)}{2m}}$ holds, so with probability at least $(1 - \delta)$, $b(\Pi) := \hat{v}(\Pi) + \frac{\gamma^d}{1-\gamma} + \frac{1-\gamma^d}{1-\gamma} \sqrt{\frac{\ln(1/\delta)}{2m}} \leq \hat{v}(\Pi) + 2\frac{\gamma^d}{1-\gamma}$ and $\nu(\Pi) := \hat{v}(\Pi) - \frac{1-\gamma^d}{1-\gamma} \sqrt{\frac{\ln(1/\delta)}{2m}} \geq \hat{v}(\Pi) - \frac{\gamma^d}{1-\gamma}$ bound $v(\Pi')$ from above and below, respectively. This choice balances the inaccuracy of estimating $v(\Pi')$ based on $v(\Pi)$ and the inaccuracy of estimating $v(\Pi)$.

Let $d^* := d^*(\epsilon, \gamma) := \lceil (\ln \frac{6}{(1-\gamma)\epsilon}) / \ln(1/\gamma) \rceil$, the smallest integer satisfying $3\frac{\gamma^{d^*}}{1-\gamma} \leq \epsilon/2$. Note that if $d(\Pi) = d^*$ for any given policy $\Pi$, then $b(\Pi) - \nu(\Pi) \leq \epsilon/2$. Because of this, it follows (see Lemma 3 in the supplementary material) that $d^*$ is the maximal length the algorithm ever has to develop a policy.

**Algorithm 1** $\mathtt{StOP}(s_0, \delta_0, \epsilon, \gamma)$

---

1: **for all** $u$ available from $x_0$ **do**         ▷ initialize
2:      $\Pi_u :=$ smallest policy with the child of $s_0$ labeled $u$
3:      $\delta_1 := (\delta_0/d^*) \cdot (K_0)^{-1}$        ▷ $d(\Pi_u) = 1$
4:      $(\nu(\Pi_u), b(\Pi_u)) := \mathtt{BoundValue}(\Pi_u, \delta_1)$
5:      $\mathrm{Active}(u) := \{\Pi_u\}$        ▷ the set of active policies that follow $u$ in $s_0$
6: **for** round t=1, 2, ... **do**
7:      **for all** $u$ available at $x_0$ **do**
8:          $\Pi^\dagger_{t,u} := \mathrm{argmax}_{\Pi \in \mathrm{Active}(u)} b(\Pi)$
9:      $\Pi^\dagger_t := \Pi^\dagger_{t,u^\dagger_t}$, where $u^\dagger_t := \mathrm{argmax}_u b(\Pi^\dagger_{t,u})$,        ▷ optimistic action and policy
10:      $\Pi^{\dagger\dagger}_t := \Pi^\dagger_{t,u^{\dagger\dagger}_t}$, where $u^{\dagger\dagger}_t := \mathrm{argmax}_{u \neq u^\dagger_t} b(\Pi^\dagger_{t,u})$,        ▷ secondary action and policy
11:      **if** $\nu(\Pi^\dagger_t) + \epsilon \geq \max_{u \neq u^\dagger_t} b(\Pi^\dagger_{t,u})$ **then**        ▷ termination criterion
12:          **return** $u^\dagger_t$
13:      **if** $d(\Pi^{\dagger\dagger}_t) \geq d(\Pi^\dagger_t)$ **then**        ▷ select the policy to evaluate
14:          $u_t := u^\dagger_t$ and $\Pi_t := \Pi^\dagger_t$
15:      **else**
16:          $u_t := u^{\dagger\dagger}_t$ and $\Pi_t := \Pi^{\dagger\dagger}_t$        ▷ action and policy to explore
17:      $\mathrm{Active}(u_t) := \mathrm{Active}(u_t) \setminus \{\Pi_t\}$
18:      $\delta := (\delta_0/d^*) \cdot \prod_{\ell=0}^{d(\Pi_t)-1} (K_\ell)^{-N^\ell}$        ▷ $\prod_{\ell=0}^{d-1}(K_\ell)^{N^\ell} =$ # of policies of depth at most $d$
19:      **for all** child policy $\Pi'$ of $\Pi_t$ **do**
20:          $(\nu(\Pi), b(\Pi)) := \mathtt{BoundValue}(\Pi', \delta)$
21:          $\mathrm{Active}(u_t) := \mathrm{Active}(u_t) \cup \{\Pi'\}$

---

### 2.2.2 Samples and sample trees

Algorithm $\mathtt{StOP}$ aims to aggressively reuse every sample for each transition node and every sample for each state-action pair, in order to keep the sample complexity as low as possible. Each time the value of a partial policy is evaluated, all samples that are available for any part of it from previous rounds are reused. That is, if $m$ trajectories are necessary for assessing the value of some policy $\Pi$, and there are $m'$ complete trajectories available and $m''$ that end at some inner node of $\Pi$, then $\mathtt{StOP}$ (more precisely, another algorithm, $\mathtt{Sample}$, called from $\mathtt{StOP}$) samples rewards (using $\mathtt{SampleReward}$) and transitions ($\mathtt{SampleTransition}$) to generate continuations for the $m''$ incomplete trajectories and to generate $(m - m' - m'')$ new trajectories, as described in Section 2.1, where

- $\mathtt{SampleReward}(s)$ for some action node $s$ samples a reward from the distribution $R(x, u, \cdot)$, where $u$ is the label of the parent of $s$ and $x$ is the label of the grandparent of $s$, and

- $\mathtt{SampleTransition}(s)$ for some transition node $s$ samples a next state from the distribution $P(x, u, \cdot)$, where $u$ is the label of $s$ and $x$ is the label of the parent of $s$.

To compensate for the sharing of the samples, the confidences of the estimates are increased, so that with probability at least $(1 - \delta_0)$, all of them are valid[6]. The samples are organized as a collection of sample trees, where a *sample tree* $\mathcal{T}$ is a (finite) subtree of $\mathbf{\Pi}^\infty$ with the property that each transition node has exactly one child, and that each action node $s$ is associated with some reward $r^{\mathcal{T}}(s)$. Note that the intersection of a policy $\Pi$ and a sample tree $\mathcal{T}$ is always a path. Denote this path by $\tau(\mathcal{T}, \Pi)$ and note that it necessarily starts from the root and ends either in a leaf or in an internal node of $\Pi$. In the former case, this path can be interpreted as a complete trajectory for $\Pi$, and in the latter case, as an initial segment. Accordingly, when the value of a new policy $\Pi$ needs to be estimated/bounded, it is computed as $\hat{v}(\Pi) := \frac{1}{m} \sum_{i=1}^m v(\tau(\mathcal{T}_i, \Pi))$ (see Algorithm 2: $\mathtt{BoundValue}$), where $\mathcal{T}_1, \ldots, \mathcal{T}_m$ are sample trees constructed by the algorithm. For terseness, these are considered to be global variables, and are constructed and maintained using algorithm $\mathtt{Sample}$ (Algorithm 3).

**Algorithm 2** `BoundValue`$(\Pi, \delta)$

---

**Ensure:** with probability at least $(1 - \delta)$, interval $[\nu(\Pi), b(\Pi)]$ contains $v(\Pi)$

1: $m := \left\lceil \frac{\ln(1/\delta)}{2} \left( \frac{1 - \gamma^{d(\Pi)}}{\gamma^{d(\Pi)}} \right)^2 \right\rceil$

2: `Sample`$(\Pi, s_0, m)$           $\triangleright$ Ensure that at least $m$ trajectories exist for $\Pi$

3: $\hat{v}(\Pi) := \frac{1}{m} \sum_{i=1}^{m} v(\tau(\mathcal{T}_i, \Pi))$          $\triangleright$ empirical estimate of $v(\Pi)$

4: $\nu(\Pi) := \hat{v}(\Pi) - \frac{1 - \gamma^{d(\Pi)}}{1 - \gamma} \sqrt{\frac{\ln(1/\delta)}{2m}}$          $\triangleright$ Hoeffding bound

5: $b(\Pi) := \hat{v}(\Pi) + \frac{\gamma^{d(\Pi)}}{1 - \gamma} + \frac{1 - \gamma^{d(\Pi)}}{1 - \gamma} \sqrt{\frac{\ln(1/\delta)}{2m}}$          $\triangleright$ ... and (2)

6: **return** $(\nu(\Pi), b(\Pi))$

---

**Algorithm 3** `Sample`$(\Pi, s, m)$

---

**Ensure:** there are $m$ sample trees $\mathcal{T}_1, \ldots, \mathcal{T}_m$ that contain a complete trajectory for $\Pi$ (i.e. $\tau(\mathcal{T}_i, \Pi)$ ends in a leaf of $\Pi$ for $i = 1, \ldots, m$)

1: **for** $i := 1, \ldots, m$ **do**
2:      **if** sample tree $\mathcal{T}_i$ does not yet exist **then**
3:          let $\mathcal{T}_i$ be a new sample tree of depth 0
4:      let $s$ be the last node of $\tau(\mathcal{T}_i, \Pi)$          $\triangleright$ $s$ is an action node
5:      **while** $s$ is not a leaf of $\Pi$ **do**
6:          let $s'$ be the child of $s$ in $\Pi$ and add it to $\mathcal{T}$ as a new child of $s$
7:          $s'' := $ `SampleTransition`$(s')$,          $\triangleright$ $s'$ is a transition node
8:          add $s''$ to $\mathcal{T}$ as a new child of $s'$
9:          $s := s''$
10:         $r^{\mathcal{T}}(s'') := $ `SampleReward`$(s'')$

---

## 3 Analysis

Recall that $v^*$ denotes the maximal value of any (possibly infinite) policy tree. The following theorem formalizes the consistency result for StOP (see the proof in Section C).

**Theorem 1.** *With probability at least $(1 - \delta_0)$, `StOP` returns an action with value at least $v^* - \epsilon$.*

Before stating the sample complexity result, some further notation needs to be introduced.

Let $u^*$ denote an optimal action available at state $x_0$. That is, $v(u^*) = v^*$. Define for $u \neq u^*$

$$\mathcal{P}_u^{\epsilon} := \left\{ \Pi \ : \ \Pi \text{ follows } u \text{ from } s_0 \text{ and } v(\Pi) + 3\frac{\gamma^{d(\Pi)}}{1 - \gamma} \geq v^* - 3\frac{\gamma^{d(\Pi)}}{1 - \gamma} + \epsilon \right\},$$

and also define

$$\mathcal{P}_{u^*}^{\epsilon} := \left\{ \Pi \ : \ \Pi \text{ follows } u^* \text{ from } s_0, \ v(\Pi) + 3\frac{\gamma^{d(\Pi)}}{1 - \gamma} \geq v^* \text{ and } v(\Pi) - 6\frac{\gamma^{d(\Pi)}}{1 - \gamma} + \epsilon \leq \max_{u \neq u^*} v(u) \right\}.$$

Then $\mathcal{P}^{\epsilon} := \mathcal{P}_{u^*}^{\epsilon} \cup \bigcup_{u \neq u^*} \mathcal{P}_u^{\epsilon}$ is the set of "important" policies that potentially need to be evaluated in order to determine an $\epsilon$-optimal action. (See also Lemma 8 in the supplementary material.)

Let now $p(s)$ denote the product of the probabilities of the transitions on the path from $s_0$ to $s$. That is, for any policy tree $\Pi$ containing $s$, a trajectory for $\Pi$ goes through $s$ with probability $p(s)$. When estimating the value of some policy $\Pi$ of depth $d$, the expected number of trajectories going through some nodes $s$ of it is $p(s)m(d, \delta_d)$. The sample complexity therefore has to take into consideration for each node $s$ (at least for the ones with "high" $p(s)$ value) the maximum $\ell(s) = \max\{d(\Pi) : \Pi \in \mathcal{P}^{\epsilon} \text{ contains } s\}$ of the depth of the relevant policies it is included in. Therefore, the expected number of trajectories going through $s$ in a given run of `StOP` is

$$p(s) \cdot m(\ell(s), \delta_{\ell(s)}) = p(s) \left\lceil \frac{\ln(1/\delta_{\ell(s)})}{2} \left( \frac{1 - \gamma^{\ell(s)}}{\gamma^{\ell(s)}} \right)^2 \right\rceil \tag{3}$$

If (3) is "large" for some $s$, it can be used to deduce a high confidence upper bound on the number of times $s$ gets sampled. To this end, let $\mathcal{S}^{\epsilon}$ denote the set of nodes of the trees in $\mathcal{P}^{\epsilon}$, let $\mathcal{N}^{\epsilon}$ denote the

smallest positive integer $\mathcal{N}$ satisfying $\mathcal{N} \geq \left| \left\{ s \in \mathcal{S}^\epsilon : p(s) \cdot m(\ell(s), \delta_{\ell(s)}) \geq (8/3) \ln(2\mathcal{N}/\delta_0) \right\} \right|$ (obviously $\mathcal{N}^\epsilon \leq |\mathcal{S}^\epsilon|$), and define

$$\mathcal{S}^{\epsilon,*} \coloneqq \left\{ s \in \mathcal{S}^\epsilon : p(s) \cdot m(\ell(s), \delta_{\ell(s)}) \geq (8/3) \ln(2\mathcal{N}^\epsilon/\delta_0) \right\}.$$

$\mathcal{S}^\epsilon$ is the set of "important" nodes ($\mathcal{P}^\epsilon$ is the set of "important" policies), and $\mathcal{S}^{\epsilon,*}$ consists of the important nodes that, with high probability, are not sampled more than twice as often as expected. (This high probability is $1 - \frac{\delta_0}{2\mathcal{N}^\epsilon}$ according to the Bernstein bound, so these upper bounds hold jointly with probability at least $(1 - \frac{\delta_0}{2})$, as $\mathcal{N}^\epsilon = |\mathcal{S}^{\epsilon,*}|$. See also Appendix D.)

For $s' \in \mathcal{S}^\epsilon \setminus \mathcal{S}^{\epsilon,*}$, the number of times $s'$ gets sampled has a variance that is too high compared to its expected value (3), so in this case, a different approach is needed in order to derive high confidence upper bounds. To this end, for a transition node $s$, let $p^\circ(s) \coloneqq p^\circ(s, \epsilon) \coloneqq \sum \{ p(s') : s'$ is a child of $s$ with $p(s') \cdot m(\ell(s'), \delta_{\ell(s')}) < (8/3) \ln(2\mathcal{N}^\epsilon/\delta_0) \}$, and define

$$B(s) \coloneqq B(s, \epsilon) \coloneqq \begin{cases} 0, & \text{if } p^\circ(s) \leq \frac{\delta}{2\mathcal{N}^\epsilon m(\ell(s), \delta_{\ell(s)})} \\ \max(6 \ln(\frac{2\mathcal{N}^\epsilon}{\delta_0}), 2p^\circ(s)m(\ell(s), \delta_{\ell(s)})) & \text{otherwise.} \end{cases}$$

As it will be shown in the proof of Theorem 2 (in Section D), this is a high confidence upper bound on the number of trajectories that go through some child $s' \in \mathcal{S}^\epsilon \setminus \mathcal{S}^{\epsilon,*}$ of some $s' \in \mathcal{S}^{\epsilon,*}$.

**Theorem 2.** *With probability at least $(1 - 2\delta)$, StOP outputs a policy of value at least $(v^* - \epsilon)$, after generating at most $\sum_{s \in \mathcal{S}^{\epsilon,*}} \left( 2p(s)m(\ell(s), \delta_{\ell(s)}) + B(s) \sum_{d=d(s)+1}^{\ell(s)} \prod_{\ell=d(s)+1}^{d} K_\ell \right)$ samples, where $d(s) = \min\{ d(\Pi) : s$ appears in policy $\Pi \}$ is the depth of node $s$.*

Finally, the bound discussed in Section 1 is obtained by setting $\kappa \coloneqq \limsup_{\epsilon \to 0} \max(\kappa_1, \kappa_2)$, where $\kappa_1 \coloneqq \kappa_1(\epsilon, \delta_0, \gamma) \coloneqq \left( \sum_{s \in \mathcal{S}^{\epsilon,*}} \frac{\epsilon^2 (1-\gamma)^2}{\ln(1/\delta_0)} 2p(s)m(\ell(s), \delta_{\ell(s)}) \right)^{1/d^*}$ and $\kappa_2 \coloneqq \kappa_2(\epsilon, \delta_0, \gamma) \coloneqq \left( \frac{\epsilon^2 (1-\gamma)^2}{\ln(1/\delta_0)} \sum_{s \in \mathcal{S}^{\epsilon,*}} B(s) \sum_{d=d(s)}^{\ell(s)} \prod_{\ell=d(s)}^{d} K_\ell \right)^{1/d^*}$.

## 4  Efficiency

StOP, as presented in Algorithm 1, is not efficiently executable. First of all, whenever it evaluates an optimistic policy, it enumerates all its child policies, which typically has exponential time complexity. Besides that, the sample trees are also treated in an inefficient way. An efficient version of StOP with all these issues fixed is presented in Appendix F of the supplementary material.

## 5  Concluding remarks

In this work, we have presented and analyzed our algorithm, StOP. To the best of our knowledge, StOP is currently the only algorithm for optimal (i.e. closed loop) online planning with a generative model that provably benefits from local structure both in reward as well as in transition probabilities. It assumes no knowledge about this structure other than access to the generative model, and does not impose any restrictions on the system dynamics.

One should note though that the current version of StOP does not support domains with infinite $N$. The sparse sampling algorithm in [14] can easily handle such problems (at the cost of a non-polynomial (in $1/\epsilon$) sample complexity), however, StOP has much better sample complexity in case of finite $N$. An interesting problem for future research is to design adaptive planning algorithms with sample complexity independent of $N$ ([21] presents such an algorithm, but the complexity bound provided there is the same as the one in [14]).

**Acknowledgments**

This work was supported by the French Ministry of Higher Education and Research, and by the European Community's Seventh Framework Programme (FP7/2007-2013) under grant agreement n$^o$ 270327 (project CompLACS). Author two would like to acknowledge the support of the BMBF project ALICE (01IB10003B).

## Footnotes

[1] A similar planning approach has been considered in the control literature, such as the model-predictive control [6] or in the AI community, such as the $A^*$ heuristic search [19] and the $AO^*$ variant [12].

[2] A problem-independent lower bound for the sample complexity, of order $(1/\epsilon)^{1/\log(1/\gamma)}$, is provided too.

[3]We emphasize the dependence on $\delta$ here since we want to compare this high-probability bound to the deterministic bound of OP-MDP.

[4]Note that leaves are, by definition, always action nodes.

[5]This approach of using secondary actions is based on the UGapE algorithm [10].

[6]In particular, the confidence is set to $1 - \delta_{d(\Pi)}$ for policy $\Pi$, where $\delta_d = (\delta_0/d^*) \prod_{\ell=0}^{d-1} K_\ell^{-N^\ell}$ is $\delta_0$ divided by the number of policies of depth at most $d$, and by the largest possible depth—see section 2.2.1.

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
