[Supplementary Material]

# A  Illustration of the StOP algorithm

(a) Iteration 1

(b) Iteration 2

(c) Iteration 3

(d) Iteration 4

(d) Iteration 5

Figure 1: Illustration of the StOP algorithm with $K = N = 2$. Black dots and arrow heads represent action nodes and transition nodes, respectively. Lines represent transitions to action nodes. The numbers given show the number of samples allocated to a node or transition. For example, in Iteration 1, the procedure Sample has allocated 6 samples to each action. The optimistic policy $\Pi^\dagger$ is selected (Step 11 of StOP), shown by the filled arrows. In iteration 2, the leaves of the optimistic policy are expanded, and Sample generates more samples along the new policies. The new optimistic policy is computed. The same process is repeated in later iterations. Note that the same samples are used to evaluate many policies, and that the leaves of the optimistic policy in Iteration 4 are not all leaves of the whole tree.

# B  Chernoff-Hoeffding and Bernstein bounds

This section provides a quick overview of the specific concentration inequalities that are used to obtain high confidence bounds on the values of the policies. The first one is the Hoeffding bound (Corollary A.1 in [7]). It implies that for any given random variable that takes values in the interval $[0, a]$ and has expected value $p$, the average $p_m$ of $m$ independent samples satisfy

$$\mathbb{P}\left[\hat{p}_m \leq p + a\sqrt{\tfrac{\ln(1/\delta)}{2m}}\right] \leq \delta \text{ and } \mathbb{P}\left[\hat{p}_m \geq p - a\sqrt{\tfrac{\ln(1/\delta)}{2m}}\right] \leq \delta.$$

The second concentration inequality is the Bernstein bound (see e.g. Corollary A.3 in [7]). It implies that for any given $a > 0$ and for any given Bernoulli variable with parameter $p$, the average $p_m$ of $m$ independent samples satisfy $\mathbb{P}\left[\hat{p}_m > p + a\right] \leq \exp\left(\frac{-a^2 m}{2p + 2a/3}\right)$ and $\mathbb{P}\left[\hat{p}_m < p - a\right] \leq$

$\exp\left(\frac{-a^2 m}{2p+2a/3}\right)$. In particular, setting $a = p$, one obtains that

$$pm \geq \tfrac{8}{3}\ln(1/\delta) \Rightarrow \quad \mathbb{P}\left[\hat{p}_m > 2p\right] = \mathbb{P}\left[\hat{p}_m > p + a\right] \leq \exp\left(\tfrac{-pm}{8/3}\right) \leq \delta \ . \tag{4}$$

Similarly, setting $a = \frac{8\ln(1/\delta)}{3m}$, one obtains that

$$pm < \tfrac{8\ln(1/\delta)}{3} \Rightarrow \quad \mathbb{P}\left[\hat{p}_m > \tfrac{16\ln(1/\delta)}{3m}\right] \leq \mathbb{P}\left[\hat{p}_m > p + a\right] \leq \exp\left(\tfrac{-am}{8/3}\right) = \delta \ . \tag{5}$$

## C   Proof of the consistency result (Theorem 1)

**Lemma 3.** *There can not be an active policy of depth larger than $d^*$.*

*Proof.* For a policy with depth larger than $d^*$ to be in an active policy set, there has to be a round $t$ with $d(\Pi_t) = d^*$. This can only be the case if $d(\Pi_t^\dagger) = d^*$ or $d(\Pi_t^{\dagger\dagger}) = d^*$. However, if $d(\Pi_t^\dagger) \geq d^*$, then it holds that $\nu(\Pi_t^\dagger) + \epsilon/2 \geq b(\Pi_t^\dagger) \geq \max_{u \neq u_t^\dagger} b(\Pi_{t,u}^\dagger)$, so StOP terminates. And since the selection rule for $u_t$ implies that $\Pi^{\dagger\dagger}$ is only selected as $\Pi_t$ if $d(\Pi_t^\dagger) > d(\Pi_t^{\dagger\dagger})$, selecting it would mean $d(\Pi_t^\dagger) > d^*$, so the algorithm would terminate by the first argument. $\square$

For convenience, we restate the theorem.

**Theorem 4** (Restatement of the consistency result, Theorem 1)**.** *With probability at least $(1 - \delta_0)$, StOP returns an action with value at least $v^* - \epsilon$.*

To prove the consistency of StOP, the following guarantee of BoundValue is needed.

**Claim 5.** *With probability at least $(1 - \delta)$, BoundValue$(\Pi, \delta)$ sets $\hat{v}(\Pi)$ to some value in the interval $\left[v(\Pi) - \frac{1-\gamma^{d(\Pi)}}{1-\gamma}\sqrt{\frac{\ln(1/d)}{2m}}, \ v(\Pi) + \frac{1-\gamma^{d(\Pi)}}{1-\gamma}\sqrt{\frac{\ln(1/d)}{2m}}\right]$.*

*Proof.* As discussed in Section 2.2.2, $\tau(\mathcal{T}_i, \Pi)$ for $i = 1, \ldots, m$ can be interpreted as trajectories for $\Pi$ that are independent (because the samples are also independent of each other). Therefore, the average of their value $\hat{v}(\Pi) = (1/m)\sum_{i=1}^m v(\tau(\mathcal{T}_i, \Pi))$ is an unbiased estimate of $v(\Pi)$. According to the Hoeffding bound (recall Section 2.2.1), the accuracy of this estimate is $\frac{1-\gamma^{d(\Pi)}}{1-\gamma}\sqrt{\frac{\ln(1/d)}{2m}} \leq \frac{\gamma^{d(\Pi)}}{1-\gamma}$ with probability at least $1 - \delta$. $\square$

Based on this, it is now easy to show that the estimates used by the algorithm are all correct with high probability.

**Corollary 6.** *The event that for every round $t$ of the algorithm, for each action $u$ available at $x_0$, for each $\Pi \in \text{Active}_t(u)$, and for each descendant $\Pi'$ of $\Pi$ (allowing $\Pi' = \Pi$), the value $v(\Pi')$ of $\Pi'$ belongs to the interval $[\nu(\Pi), b(\Pi)]$, has probability at least $(1 - \delta_0)$, and implies $\nu\left(\Pi_{t,u}^\dagger\right) \leq v(u) \leq b\left(\Pi_{t,u}^\dagger\right)$.*

*Proof.* If BoundValue is ever called for some policy $\Pi$, then it is called with confidence parameter $\delta$ set to $\delta_d = (\delta_0/d^*)\prod_{\ell=1}^d K_\ell$, where $d = d(\Pi)$ is the depth of $\Pi$. Note also that $\prod_{\ell=0}^{d-1}(K_\ell)^{N^\ell}$ is the number of partial policies of depth $d$, and therefore, based on Claim 5 and Lemma 3, with probability at least $1 - \sum_{d=1}^{d^*} \delta_d \prod_{\ell=0}^{d-1}(K_\ell)^{N^\ell} = 1 - \delta_0$, for every $\Pi$ that ever belongs to the set of active policies, $v(\Pi) \in \left[\hat{v}(\Pi) - \frac{1-\gamma^{d(\Pi)}}{1-\gamma_{d\Pi}}\sqrt{\frac{\ln(1/d)}{2m}}, \hat{v}(\Pi) + \frac{1-\gamma^{d(\Pi)}}{1-\gamma_{d(\Pi)}}\sqrt{\frac{\ln(1/d)}{2m}}\right]$. The claimed result now follows from (2). $\square$

The consistency result of Theorem 1 follows immediately from Corollary 6, Lemma 3, and the termination condition of StOP.

# D  Proof of the sample complexity (Theorem 2)

For convenience, we restate the theorem.

**Theorem 7** (Restatement of the sample complexity bound, Theorem 2)**.** *With probability at least* $(1 - 2\delta)$, $\mathtt{StOP}$ *outputs a policy of value at least* $(v^* - \epsilon)$ *after generating at most*

$$\sum_{s \in \mathcal{S}^{\epsilon,*}} \left( 2p(s)m(\ell(s), \delta_{\ell(s)}) + B(s) \sum_{d=d(s)+1}^{\ell(s)} \prod_{\ell=d(s)+1}^{d} K_\ell \right) \tag{6}$$

*samples, where* $d(s) = \min\{d(\Pi) : s \text{ appears in policy } \Pi\}$ *is the depth of node s.*

For the proof we require that $\mathcal{P}^\epsilon$ does indeed contain, with high probability, all the important policies. The following lemma is essential for this.

**Lemma 8.** *Assume that for each* $t \geq 0$, *for each action available at* $x_0$, *for each policy* $\Pi \in \text{Active}_t(u)$, $\nu(\Pi) \leq v(\Pi) \leq b(\Pi)$. *Then* $\Pi_t \in \mathcal{P}_\epsilon$ *for every* $t \geq 1$ *throughout the whole run of the algorithm, except for (possibly) the last round.*

*Proof.* Note that, whenever a policy is removed from the set of active policies, it is replaced by its child policies. So, as $\Pi_{u^*} \in \text{Active}(u^*)$ initially, in every subsequent step there will be some $\Pi \in \text{Active}(u^*)$ that has a descendant policy of value $v^*$. Therefore, by the assumption of the lemma and by Corollary 6, we have $b(\Pi_{t,u^*}^\dagger) \geq v^*$, and therefore

$$b(\Pi_t^\dagger) \geq b\left(\Pi_{t,u^*}^\dagger\right) \geq v^*. \tag{7}$$

Additionally, the selection rule of $\Pi_t$ implies

$$d(\Pi_t) \leq \min\left\{ d(\Pi_t^\dagger), d(\Pi_t^{\dagger\dagger}) \right\}. \tag{8}$$

For any $u \neq u^*$ this implies that, whenever $\Pi_t = \Pi_{t,u}^\dagger$ and the termination criterion is not met,

$$
\begin{aligned}
v(\Pi_t) + 3\frac{\gamma^{d(\Pi_t)}}{1-\gamma} - \epsilon &\geq \nu(\Pi_t) + 3\frac{\gamma^{d(\Pi_t)}}{1-\gamma} - \epsilon && \text{by the assumption} \\
&\geq b(\Pi_t) - \epsilon && \text{by the definition of } b \text{ and } \nu \\
&\geq \max_{u \neq u_t^\dagger} b(\Pi_{t,u}^\dagger) - \epsilon && \text{by the choice of } \Pi_t \\
&> \nu(\Pi_t^\dagger) && \text{termination criterion is not met} \\
&\geq b(\Pi_t^\dagger) - 3\frac{\gamma^{d(\Pi_t^\dagger)}}{1-\gamma} && \text{by the definition of } b \text{ and } \nu \\
&\geq v^* - 3\frac{\gamma^{d(\Pi_t^\dagger)}}{1-\gamma} && \text{by (7)} \\
&\geq v^* - 3\frac{\gamma^{d(\Pi_t)}}{1-\gamma} && \text{by (8)}
\end{aligned}
$$

Consequently $\Pi_t \in \mathcal{P}^\epsilon$.

Similarly, when $\Pi_t = \Pi_{t,u^*}^\dagger$ then $\{u_t^\dagger, u_t^{\dagger\dagger}\} = \{u^*, u'\}$ for some $u'$, and, if the termination criterion is not met, then

$$
\begin{aligned}
\max_{u \neq u^*} v(u) + 3\frac{\gamma^{d(\Pi_t)}}{1-\gamma} &\geq \max_{u \neq u^*} \nu(\Pi_{t,u}^\dagger) + 3\frac{\gamma^{d(\Pi_t)}}{1-\gamma} && \text{by the assumption} \\
&\geq \max_{u \neq u^*} \nu(\Pi_{t,u}^\dagger) + 3\frac{\gamma^{d(\Pi_{t,u'}^\dagger)}}{1-\gamma} && \text{because of (8) and } \{u_t^\dagger, u_t^{\dagger\dagger}\} = \{u^*, u'\} \\
&\geq \nu(\Pi_{t,u'}^\dagger) + 3\frac{\gamma^{d(\Pi_{t,u'}^\dagger)}}{1-\gamma} && \text{because } u' \neq u^* \\
&\geq b(\Pi_{t,u'}^\dagger) && \text{by the definition of } b \text{ and } \nu \\
&= \max_{u \neq u^*} b(\Pi_{t,u}^\dagger) && \text{because } \{u_t^\dagger, u_t^{\dagger\dagger}\} = \{u^*, u'\}
\end{aligned}
$$

$$\geq \max_{u \neq u_t^\dagger} b(\Pi_{t,u}^\dagger) \qquad\qquad \text{by the choice of } u_t^\dagger$$

$$\geq \nu(\Pi_t^\dagger) + \epsilon \qquad\qquad \text{termination criterion is not met}$$

$$\geq b(\Pi_t^\dagger) - 3\frac{\gamma^{d(\Pi_{t,u}^\dagger)}}{1-\gamma} + \epsilon \qquad\qquad \text{by the definition of } b \text{ and } \nu$$

$$\geq b(\Pi_t) - 3\frac{\gamma^{d(\Pi_{t,u}^\dagger)}}{1-\gamma} + \epsilon \qquad\qquad \text{by the choice of } \Pi_t$$

$$\geq b(\Pi_t) - 3\frac{\gamma^{d(\Pi_t)}}{1-\gamma} + \epsilon \qquad\qquad \text{by (8)}$$

$$\geq v(\Pi_t) - 3\frac{\gamma^{d(\Pi_t)}}{1-\gamma} + \epsilon \qquad\qquad \text{by the assumption}$$

This, combined with (7), implies that $\Pi_t \in \mathcal{P}_{u^*}^\epsilon$. $\qquad\square$

*Proof of Theorem 6.* In the proof it is assumed that $\Pi_t \in \mathcal{P}^\epsilon$ for every $t$ throughout the algorithm, except for (possibly) the last round. According to Lemma 8 and Corollary 6, this holds with probability at least $(1 - \delta_0)$.

The assumption implies that all rollouts generated by StOP consist of nodes that belong to $\mathcal{S}^\epsilon$. It also implies that for any node $s$ of $\Pi^\infty$, the depth of any policy $\Pi$ that includes $s$ and is evaluated by StOP is bounded by $\ell(s)$. The largest amount of samples required by such a policy is thus $m(\ell(s), \delta_{\ell(s)})$. Therefore, according to the Bernstein bound (4), for any $s \in \mathcal{S}^{\epsilon,*}$, the number of sample trees that contain $s$ is bounded from above by $2p(s)m(\ell(s), \delta_{\ell(s)})$ with probability at least $(1 - \delta_0/(2\mathcal{N}^\epsilon))$, and so this also upper bounds the number of samples that are generated for $s$.

It now remains to upper bound the number of samples that are generated for nodes in $(\mathcal{S}^\epsilon \setminus \mathcal{S}^{\epsilon,*})$. For this, first partition these nodes by forming, for each $s \in \mathcal{S}^{\epsilon,*}$, a group which consists of all the nodes that have $s$ as their lowest ancestor in $\mathcal{S}^{\epsilon,*}$. Note that the probability that a trajectory traverses through this group is $p^\circ(s)$, and therefore, according to the Bernstein bound, the number of trajectories that traverses this group is upper bounded by $B(s)$ with probability at least $(1 - \delta/(2\mathcal{N}^\epsilon))$. Indeed, if $p^\circ(s)m(\ell(s), \delta_{\ell(s)}) \geq (8/3)\ln(2\mathcal{N}^\epsilon/\delta)$, the Bernstein bound (4) guarantees the bound $2p^\circ(s)m(\ell(s), \delta_{\ell(s)})$ with confidence at least $(1 - \delta/(2\mathcal{N}^\epsilon))$, and otherwise (5) provides the bound $p^\circ(s)m(\ell(s), \delta_{\ell(s)}) + 3\ln(2\mathcal{N}^\epsilon/\delta) \leq 6\ln(2\mathcal{N}^\epsilon/\delta)$. In fact, if $p^\circ(s) \leq \delta/(2\mathcal{N}^\epsilon m(\ell(s), \delta_{\ell(s)}))$ then, from the Bernoulli inequality, with probability at least $(1 - \delta_0/(2\mathcal{N}^\epsilon))$, no trajectory traverses the group. Finally, note that a sample tree contains at most $\sum_{d=d(s)+1}^{\ell(s)} \prod_{\ell=d(s)+1}^{d} K_\ell$ samples below node $s$. $\qquad\square$

## E   Worst case bound and special cases

Before we turn to the analysis of the special cases, we discuss shortly the second term in the sample complexity bound (6).

**Claim 9.** $\sum_{s \in \mathcal{S}^{\epsilon,*}} B(s) \sum_{d=d(s)+1}^{\ell(s)} \prod_{\ell=d(s)+1}^{d} K_\ell \leq |\mathcal{S}^\epsilon \setminus \mathcal{S}^{\epsilon,*}| \cdot 6 \cdot \ln(\frac{2\mathcal{N}^\epsilon}{\delta_0})$

*Proof.* First of all, each $s \in \mathcal{S}^{\epsilon,*}$ has at least $p^\circ(s)(3/8)m(d, \delta_{\ell(s)})/\ln(2\mathcal{N}^\epsilon/\delta_0)$ children $s'$ with $p(s')m(d, \delta_{\ell(s')}) < (8/3)\ln(2\mathcal{N}^\epsilon/\delta_0)$ (note that $\ell(s) = \ell(s')$), and therefore the maximum of $6\ln(\frac{2\mathcal{N}^\epsilon}{\delta_0})$ and $2p^\circ(s)m(\ell(s), \delta_{\ell(s)})$ is upper bounded by the number of these children multiplied by $6\ln(2\mathcal{N}^\epsilon/\delta_0)$. Note also that number of nodes in $\mathcal{S}^\epsilon$ below $s'$ is at least $\sum_{d=d(s)+1}^{\ell(s)} \prod_{\ell=d(s)+1}^{d} K_\ell$. Summing up, $B(s)$ accounts at most $6\ln\frac{2\mathcal{N}^\epsilon}{\delta_0}$ to every $s' \in \mathcal{S}^\epsilon \setminus \mathcal{S}^{\epsilon,*}$ that has $s$ as its lowest ancestor in $\mathcal{S}^{\epsilon,*}$. $\qquad\square$

Now recall that $d^* = d^*(\epsilon, \gamma) = \left\lceil \frac{\ln((1-\gamma)\epsilon/6)}{\ln\gamma} \right\rceil$, and also that this implies

$$\epsilon(1-\gamma) \leq 6\gamma^{d^*-1}. \tag{9}$$

Defining

$$\kappa_1 := \kappa_1(\epsilon, \delta_0, \gamma) := \left( \sum_{s \in \mathcal{S}^{\epsilon,*}} \frac{\epsilon^2 (1-\gamma)^2}{\ln(1/\delta_0)} 2p(s) m(\ell(s), \delta_{\ell(s)}) \right)^{1/d^*}$$

$$\leq \left( \frac{\epsilon^2 (1-\gamma)^2}{\ln(1/\delta_0)} \sum_{s \in \mathcal{S}^{\epsilon,*}} p(s) \cdot \frac{1}{\gamma^{2\ell(s)}} \ln \frac{d^* \prod_{\ell=1}^{\ell(s)} (K_\ell)^{N^\ell}}{\delta_0} \right)^{1/d^*}$$

$$\leq \left( \frac{\epsilon^2 (1-\gamma)^2}{\gamma^{2d^*}} \sum_{s \in \mathcal{S}^{\epsilon,*}} p(s) \left( \ln d^* + \sum_{\ell=1}^{\ell(s)} N^\ell \ln K_\ell \right) \right)^{1/d^*}$$

$$\leq \left( \frac{6}{\gamma^2} \sum_{s \in \mathcal{S}^{\epsilon,*}} p(s) \left( \ln d^* + \sum_{\ell=1}^{\ell(s)} N^\ell \ln K_\ell \right) \right)^{1/d^*} \qquad \text{(by 9)),}$$

one obtains the bound

$$\sum_{s \in \mathcal{S}^{\epsilon,*}} 2p(s) m(\ell(s), \delta_{\ell(s)}) = \frac{\ln(1/\delta_0)}{(1-\gamma)^2 \epsilon^2} \sum_{s \in \mathcal{S}^{\epsilon,*}} \frac{\epsilon^2 (1-\gamma)^2}{\ln(1/\delta_0)} 2p(s) m(\ell(s), \delta_{\ell(s)})$$

$$= \frac{\ln(1/\delta_0)}{\epsilon^2 (1-\gamma)^2} \cdot \kappa_1^{d^*}$$

$$= \frac{\ln(1/\delta_0)}{\epsilon^2 (1-\gamma)^2} \cdot \kappa_1^{\frac{\ln((1-\gamma)\epsilon) - \ln 6}{\ln \gamma}}$$

$$= \left( \ln \tfrac{1}{\delta_0} \right) \cdot \kappa_1^{\frac{\ln 6}{\ln(1/\gamma)}} \cdot \left( \frac{1}{(1-\gamma)\epsilon} \right)^{2 + \frac{\ln \kappa_1}{\ln(1/\gamma)}}.$$

Similarly, defining

$$\kappa_2 := \kappa_2(\epsilon, \delta_0, \gamma) := \left( \frac{\epsilon^2 (1-\gamma)^2}{\ln(1/\delta_0)} \sum_{s \in \mathcal{S}^{\epsilon,*}} B(s) \sum_{d=d(s)}^{\ell(s)} \prod_{\ell=d(s)}^{d} K_\ell \right)^{1/d^*}$$

$$= \left( \frac{\epsilon^2 (1-\gamma)^2}{\ln(1/\delta_0)} \cdot |\mathcal{S}^\epsilon \setminus \mathcal{S}^{\epsilon,*}| \cdot 6 \cdot \ln\left( \frac{2|\mathcal{S}^{\epsilon,*}|}{\delta_0} \right) \right)^{1/d^*} \qquad \text{(by Claim (9))}$$

$$\leq \left( \epsilon^2 (1-\gamma)^2 \cdot |\mathcal{S}^\epsilon \setminus \mathcal{S}^{\epsilon,*}| \cdot 6 \cdot \ln(2|\mathcal{S}^{\epsilon,*}|) \right)^{1/d^*}$$

$$\leq \left( 6\gamma^{2d^* - 2} \cdot |\mathcal{S}^\epsilon \setminus \mathcal{S}^{\epsilon,*}| \cdot 6 \cdot \ln(2|\mathcal{S}^{\epsilon,*}|) \right)^{1/d^*} \qquad \text{(by (9)),}$$

one obtains the bound

$$\sum_{s \in \mathcal{S}^{\epsilon,*}} B(s) \sum_{d=d(s)}^{\ell(s)} \prod_{\ell=d(s)}^{d} K_\ell = \frac{\ln(1/\delta_0)}{\epsilon^2 (1-\gamma)^2} \cdot \kappa_2^{\frac{\ln((1-\gamma)\epsilon) - \ln 6}{\ln \gamma}} = \left( \ln \tfrac{1}{\delta_0} \right) \cdot \kappa_2^{\frac{\ln 6}{\ln(1/\gamma)}} \cdot \left( \frac{1}{(1-\gamma)\epsilon} \right)^{2 + \frac{\ln \kappa_1}{\ln(1/\gamma)}}.$$

Finally, defining $\kappa := \limsup_{\epsilon \to 0} \max(\kappa_1, \kappa_2)$, one obtains the following sample complexity bound.

**Theorem 10.** *Sample complexity (6) is upper bounded by* $\left( \ln \tfrac{1}{\delta_0} \right) \cdot C(\kappa, \gamma) \cdot \left( \frac{1}{(1-\gamma)\epsilon} \right)^{2 + \frac{\ln \kappa}{\ln(1/\gamma)}}$, *where* $C(\kappa, \gamma) := 2\kappa^{\frac{\ln 6}{\ln(1/\gamma)}}$.

### E.1 Worst case

If $K_\ell = K > 1$ for each $\ell > 0$ then $\sum_{s \in \mathcal{S}^{\epsilon,*}} p(s) = \sum_{s \in \mathcal{S}^\epsilon} p(s) \leq K^{d^*}$, so

$$\kappa_1 \leq \left( \frac{6 \left( \ln d^* + N^{d^*} d^* \ln K \right)}{\gamma^2} \sum_{s \in \mathcal{S}^{\epsilon,*}} p(s) \right)^{1/d^*} \leq \left( \frac{6 \left( \ln d^* + N^{d^*} d^* \ln K \right) K^{d^*}}{\gamma^2} \right)^{1/d^*}.$$

Therefore, $\limsup_{\epsilon\to 0}\kappa_1 \le KN$. Similarly, noting that $|\mathcal{S}^\epsilon| \le (NK)^{d^*}$,

$$\kappa_2 \le \left(\gamma^{2d^*-2}\cdot(NK)^{d^*}\cdot 6\cdot d^*\ln(NK)\right)^{1/d^*},$$

which implies $\limsup_{\epsilon\to 0}\kappa_2 \le \gamma^2 KN$.

## E.2 Case $K_0 > 1, K_\ell = 1$ **for all** $\ell \ge 1$

In this case

$$\sum_{s\in\mathcal{S}^{\epsilon,*}} p(s) \le d^*K, \qquad (10)$$

and so

$$\kappa_1 \le \left(\frac{6}{\gamma^2}\sum_{s\in\mathcal{S}^{\epsilon,*}} p(s)\left(\ln d^* + N\ln K\right)\right)^{1/d^*} = \left(\frac{6}{\gamma^2}\left(\ln d^* + N\ln K\right)d^*K\right)^{1/d^*},$$

which implies $\limsup_{\epsilon\to 0}\kappa_1 \le 1$.

To bound $\kappa_2$, note that $p^\circ(s) \le p(s)$ for all $s$ and that $\sum_{d=1}^{d^*}\prod_{\ell=d}^{d^*} K_\ell = 1$, which implies

$$\kappa_2 \le \left(\frac{\epsilon^2(1-\gamma)^2}{\ln(1/\delta_0)}\sum_{s\in\mathcal{S}^\epsilon}\left(2p(s)m(\ell(s),\delta_{\ell(s)}) + 6\ln(\frac{2\mathcal{N}^\epsilon}{\delta_0})\right)\right)^{1/d^*}$$

$$\le \left(\kappa_1^{d^*} + \frac{\epsilon^2(1-\gamma)^2}{\ln(1/\delta_0)}\cdot|\mathcal{S}^{\epsilon,*}|\cdot 6\ln(\frac{2\mathcal{N}^\epsilon}{\delta_0})\right)^{1/d^*}.$$

By (10) and the definition of $\mathcal{S}^{\epsilon,*}$, the restriction that $K_\ell = 1$ for all $\ell > 1$ implies

$$|\mathcal{S}^{\epsilon,*}| \le K\cdot d^*\frac{3m(d^*,\delta_{d^*})}{8\ln(2\mathcal{N}^\epsilon/\delta_0)} \le K\cdot d^*\frac{3N\ln(d^*K/\delta_0)}{16\gamma^{2d^*}\ln(1/\delta_0)}.$$

Therefore, recalling also (9),

$$\kappa_2 \le \left(\kappa_1 + \frac{\gamma^{2d^*-2}}{\ln(1/\delta_0)}K\cdot d^*\frac{3N\ln(d^*K/\delta_0)}{16\gamma^{2d^*}\ln(1/\delta_0)}6d^*\ln(\frac{KN}{\delta_0})\right)^{1/d^*}$$

$$= \left(\kappa_1 + \frac{(d^*)^2 2NK}{\gamma^2}\frac{\ln(d^*K/\delta_0)\ln(KN/\delta_0)}{\ln^2(1/\delta_0)}\right)^{1/d^*}.$$

Consequently, $\limsup_{\epsilon\to 1}\kappa_2 \le 1$ as well.

## E.3 Bandit case

Again $K_0 > 1, K_\ell = 1$ for all $\ell \ge 1$, but it is also assumed that $N = 1$ and that all the rewards in the same branch are equal (they can be different though between different branches). Then, directly from (6), one easily deduces the bound $O\left(\left(\ln\frac{d^*}{\delta_0}\right)\sum_{u\ne u^*}\left(\frac{1}{(1-\gamma)(v^*-v(u)+\epsilon)}\right)^{-2}\right)$.

## E.4 Deterministic MDPs

In case $N = 1$ and $K_\ell = K > 1$ for $\ell \ge 0$, we have $\kappa_1 \le \left(\frac{6}{\gamma^2}\cdot K^{d^*}\cdot(\ln d^* + d^*\cdot\ln K)\right)^{1/d^*}$, so $\limsup_{\epsilon\to 0}\kappa_1 \le K$. Additionally, $\kappa_2 = 0$, since in this case $p(s) = 1$ for each node $s$.

Assume now some structure in the rewards: for every action $u$ on exactly one path in $\Pi^\infty$, the rewards are 1; everywhere else they are 0. Then, nodes with depth at least $\log(5)/\log(1/\gamma)$ bigger than their lowest nonzero-reward ancestor do not appear in $\mathcal{S}^\epsilon$. Therefore,

$$\kappa_1 \le \left(\frac{\epsilon^2(1-\gamma)^2}{\ln(1/\delta_0)}\cdot K\cdot\sum_{d=1}^{d^*} K^{\log(5)/\log(1/\gamma)}m(d,\delta_d)\right)^{1/d^*}$$

$$\le \left(\frac{3}{\gamma^2\ln(1/\delta_0)}\cdot d^*K^{1+\log(5)/\log(1/\gamma)}\ln\frac{d^*K^{d^*}}{\delta_0}\right)^{1/d^*},$$

and so $\limsup_{\epsilon\to 0}\kappa_1 = 1$.

# F Efficient version of StOP

This section is devoted to fix all the time-efficiency issues in the previous version of the algorithm. The primary task here is to find a way to solve both the policy evaluation and the construction of the optimistic policies efficiently.

With some abuse of notation, let $\text{Active}_t$ denote the set in round $t$ consisting of policies $\Pi$ for which rollout $\tau(\Pi, \mathcal{T}_i)$ has length $d(\Pi)$ for $1 \leq i \leq m(d(\Pi), \delta_{d(\Pi)})$, and, at the same time, for some child policy $\Pi'$ of $\Pi$ some rollout $\tau(\Pi, \mathcal{T}_i)$ for $1 \leq i \leq m(d(\Pi), \delta_{d(\Pi)})$ has length less than $d(\Pi')$.

## F.1 Evaluating the children of $\Pi_t$

The first problem to solve is to maintain the sample trees without actually going through all the child policies of $\Pi_t$.

To this end, define first $m_d(s)$ as the number of times $s$ appears in sample trees $\mathcal{T}_1, \mathcal{T}_2, \ldots, \mathcal{T}_{m(d,\delta_d)}$ in the current round. Similarly, let $\hat{r}_d(s)$ denote the average of the rewards for $s$ in $\mathcal{T}_1, \mathcal{T}_2, \ldots, \mathcal{T}_{m(d,\delta_d)}$ at the current round. These values are easily updated using a simple recursion rule applied in algorithm `Sample-eff`.

**Claim 11.** *Executing* `Sample-eff`$(\Pi, s, m)$ *ensures that* $\tau(\mathcal{T}_i, \Pi)$ *has length* $d(\Pi)$ *(i.e., has full length) for* $i = 1, 2, \ldots, m$, *and runs in time* $O(m \cdot d(\Pi))$.

As the next step note that, if the first $K_d$ child policies of $\Pi_t$ which `StOP` picks to evaluate in round $t$ (where $d = d(\Pi_t)$) do not share any leaves, then `BoundValue` will not call `SampleTransition` or `SampleReward` for any other children of $\Pi_t$. The reason for this is that the the first $K_d$ trees include all the nodes that appear in any child policy of $\Pi_t$.

The above argument shows that the evaluation of a policy $\Pi$ in `StOP-eff` and in `StOP` are essentially equivalent.

## F.2 Constructing the optimistic policies

Note that, in round $t$, for any $\Pi \in \cup_{t' \leq t} \text{Active}$ it holds that

$$\hat{v}(\Pi) = \sum_{s \in \Pi} \gamma^{d(s)} \cdot m_d(s) \cdot \hat{r}_d(s) \ .$$

Additionally, as $b(\Pi) = \hat{v}(\Pi) + 2\frac{\gamma^{d(\Pi)}}{1-\gamma}$, it holds for any two policies $\Pi$ and $\Pi'$ of the same depth that

$$b(\Pi) > b(\Pi') \quad \Leftrightarrow \quad \hat{v}(\Pi) > \hat{v}(\Pi') \ .$$

It is therefore easy to compute the value of any active policy, and also to decide which of two policies is better. However, it is less obvious how to construct the optimistic policies efficiently.

**Theorem 12.** *For any action $u$ accessible from $x_0$, and any round $t$,* `ValueTr`$(s_u)$ *returns* $\Pi_{t,u}^{\dagger}$, *where $s_u$ is the child of the root labeled $u$.*

*Proof.* Let $a_d(s) = a_{d,t}(s)$ be the indicator that, for some $1 \leq i \leq m(d, \delta_d)$, sample tree $\mathcal{T}_i$ has a leaf below $s$ with $d(s) = d$ at iteration $t$. Note that for action node $s$, $a_d(s)$ must be set to 1 if $m_{d(s)}(s) > 0$ and $m_{d(s)+1}(s') = 0$ for some child $s'$ of $s$, otherwise it must be set to 0. For node $s$ of depth $d(s) < d$, $a_{t,d}(s)$ can be computed based on the simple recursion rule $a_d(s) := \max_{s' \text{ child of } s} a_d(s')$.

Equivalently, $a_{d,t}(s)$ indicates that, for some policy $\Pi$ of depth $d$ containing $s$, rollout $\tau(\mathcal{T}_i, \Pi)$ has length $d$ (i.e., full length) for $i = 1, \ldots, m(d, \delta_d)$, but for some child policy $\Pi'$ of $\Pi$ and for some $1 \leq i \leq m(d, \delta_d)$ rollout $\tau(\Pi', \mathcal{T}_i)$ *goes through* $s$ and has length at most $d$ (instead of $d + 1$, which would be the maximal possible). On one hand, the extra requirement about the rollout going through $s$ makes a distinction between $a_{d,t}(s)$ and the indicator that $s$ belongs to some policy in $\text{Active}_t$, but, at the same time, this is the distinction that makes it easy to compute it efficiently with the recursive rule described above. This is the key insight that is used in constructing the optimistic policies efficiently, too.

Now, consider, for each node $s$ the policies in $\cup_{t' \leq t}\text{Active}_{t'}$ with $d(\Pi) = d$, and denote by $\Pi_{t,d}^{\text{comp}}(s)$ the one that has the largest cumulative reward below $s$ in the first $m(d, \delta_d)$ sample trees. Denote this cumulative reward by $\hat{v}^{\text{comp}}(s)$, and note that it can be computed recursively by

- setting it to $\hat{r}_d(s)$ for each action node $s$ with $d(s) = d$,

- setting it to $\max_{s' \text{ children of } s} \hat{v}_d^{\text{comp}}(s')$ for all action nodes with $d(s) < d$, and

- setting it to $\hat{v}_d^{\text{compl}}(s) := \gamma \sum_{s': \text{child of } s}(m_d(s') \cdot \hat{v}_d^{\text{compl}}(s'))$ for a transition node $s$ with $d(s) \leq d$.

Finally, consider, for a node $s$, those policies in $\cup_{t' \leq t}\text{Active}_{t'}$ that satisfy

- $d(\Pi) = d$

- rollout $\tau(\mathcal{T}_i, \Pi)$ has length $d$ (i.e., full length) for $i = 1, \ldots, m(d, \delta_d)$,

- for some child policy $\Pi'$ of $\Pi$ and for some $1 \leq i \leq m(d, \delta_d)$ rollout $\tau(\Pi', \mathcal{T}_i)$ *goes through $s$ and has length $d$ too* (instead of $d + 1$).

Denote by $\Pi_{t,d}^{\text{inc}}(s)$ the one that has the largest cumulative reward below $s$ in the first $m(d, \delta_d)$ sample trees, and by $\hat{v}_d^{\text{inc}}$ this cumulative reward. This value can also be computed efficiently using recursion:

- $\hat{v}_d^{\text{inc}}(s) := \hat{r}_d(s)$ for a transition node $s$ with $d(s) = d$

- $\hat{v}_d^{\text{inc}}(s) := \max_{s' \text{ children of } s \text{ with } a_d(s)=1} \hat{v}_d^{\text{inc}}(s')$ for a transition node $s$ with $d(s) < d$, and

- 
$$\hat{v}_d^{\text{inc}}(s) := \gamma \max_{s': \text{ child of } s \text{ with } a_d(s')=1} \left( m_d(s') \cdot \hat{v}_d^{\text{inc}}(s') + \sum_{s'' \neq s' \text{ child of } s} (m_d(s'') \cdot \hat{v}_d^{\text{compl}}(s'')) \right)$$

  for an action node $s$ with $d(s) \leq d$

The claim of the theorem follows by noting that, for any child node $s$ of the root, $\Pi_{t,d}^{\text{inc}}(s) = \Pi_{t,u}^{\dagger}$, where $u$ is the label of $s$. $\qquad\square$

In order to simplify the pseudocode, the construction of the optimistic policies is not implemented. Nevertheless, they can be easily obtained similar to how the values $\hat{v}_d^{\text{comp}}(s)$ and $\hat{v}_d^{\text{inc}}(s)$ are computed.

Finally, note that in a given step $t$, only those values that belong to the nodes of policy $\Pi_t$ require updating. Making use of this, an even more significant speed-up is possible.

**Algorithm 4** $\texttt{StOP-eff}(s_0, \delta_0, \epsilon, \gamma)$

1: **for all** $u$ available from $x_0$ **do**            $\triangleright$ Initialize
2:      $\Pi :=$ smallest policy with the child $s_u$ of $s_0$ labeled $u$
3:      $\delta_1 := (\delta_0/d^*) \cdot (K_0)^{-1}$            $\triangleright d(\Pi) = 1$
4:      $\texttt{Sample}(\Pi, s_u, m(1, \delta_1))$
5: $t := 1$
6: **for** round $t = 1, 2, \dots$ **do**
7:      **for all** $u$ available at $x_0$ **do**
8:          $\texttt{ValueTr}(s_u)$
9:          $\Pi_{t,u}^{\dagger} := \mathrm{argmax}_{\Pi \in \mathrm{Active}(u)} \, b(\Pi)$
10:      $\Pi_t^{\dagger} := \Pi_{t,u_t^{\dagger}}^{\dagger}$, where $u_t^{\dagger} := \mathrm{argmax}_u \, b(\Pi_{t,u}^{\dagger})$         $\triangleright$ optimistic policy and action
11:      $\Pi_t^{\dagger\dagger} := \Pi_{t,u_t^{\dagger\dagger}}^{\dagger}$, where $u_t^{\dagger\dagger} := \mathrm{argmax}_{u \neq u_t^{\dagger}} \, b(\Pi_{t,u}^{\dagger})$         $\triangleright$ secondary policy and action
12:      **if** $\nu(\Pi_t^{\dagger}) + \epsilon \geq \max_{u \neq u_t^{\dagger}} b(\Pi_{t,u}^{\dagger})$ **then**         $\triangleright$ termination criterion
13:          **return** $u_t^{\dagger}$
14:      **if** $d(\Pi_t^{\dagger\dagger}) \geq (\Pi_t^{\dagger})$ **then**         $\triangleright$ choose action and policy to explore
15:          $u_t := u_t^{\dagger}$ and $\Pi_t := \Pi_t^{\dagger}$
16:      **else**
17:          $u_t := u_t^{\dagger\dagger}$ and $\Pi_t := \Pi_t^{\dagger\dagger}$
18:      set $d_t := d(\Pi_t)$
19:      $\delta := (\delta_0/d^*) \cdot \prod_{\ell=0}^{d_t-1}(K_\ell)^{-N^\ell}$     $\triangleright$ the # of policies of depth at most $d$ is $\prod_{\ell=0}^{d-1}(K_\ell)^{N^\ell}$
20:      **for** each of the $K_{d_t}$ action $u$ **do**
21:          let $\Pi_{t,u}$ be the policy children of $\Pi$ that follow action $u$ from each leaf of $\Pi$
22:          set $a_{d_t}(s) := 1$ for each node $s$ of $\Pi_{t,u}$ that is not in $\Pi_t$
23:          $\texttt{Sample}\left(\Pi_{t,i}, \, s_{u_t}, \, m(d_t + 1, \delta_{d_t+1})\right)$
24:      $t := t + 1$

---

**Algorithm 5** $\texttt{Sample-eff}(\Pi, s, m)$

1: **if** $s$ is a leaf of $\Pi$ **then return**
2: let $s'$ be the child node of $s$ in $\Pi$
3: **while** $m_{d(\Pi)}(s') < m$          $\triangleright$ make sure that $s$ has at least $m$ samples **do**
4:      $m_{d(\Pi)}(s') := m_{d(\Pi)}(s') + 1$
5:      $s'' := \texttt{SampleTransition}(s')$
6:      $\hat{r}_{d(\Pi)}(s'') := \frac{\hat{r}_{d(\Pi)}(s'') \cdot m_{d(\Pi)}(s'') + \texttt{SampleReward}(s'')}{1 + m_{d(\Pi)}(s'')}$
7:      $m_{d(\Pi)}(s'') := m_{d(\Pi)}(s'') + 1$
8: **for all** grandchildren $s''$ of $s$ **do** $\triangleright$ ensure that all rollouts going through $s$ have full length in $\Pi$
9:      $\texttt{Sample-eff}(\Pi, s'', m_{d(\Pi)}(s''))$

---

**Algorithm 6** $\texttt{ValueTr}(s)$

1: $a_d(s) = 0$
2: **for all** children $s'$ of $s$ with $\max_{d=d(s'),\dots,d^*} m_d(s') > 0$ **do**
3:      $\texttt{ValueAc}(s')$
4: **for all** $d := d(s) + 1, \dots, d^*$ with $m_d(s) > 0$ **do**
5:      $\hat{v}_d^{\mathrm{compl}}(s) := \gamma \sum_{s': \text{ child of } s}(m_d(s') \cdot \hat{v}_d^{\mathrm{compl}}(s'))$
6:      $a_d(s) := \max_{s' \text{ child of } s} a_d(s')$
7:      $\hat{v}_d^{\mathrm{inc}}(s) := \gamma \max_{s': \text{ child of } s \text{ with } a_d(s')=1} \Big( m_d(s') \cdot \hat{v}_d^{\mathrm{inc}}(s')$
8:                             $+ \sum_{s'' \neq s' \text{ child of } s}(m_d(s'') \cdot \hat{v}_d^{\mathrm{compl}}(s'')) \Big)$

**Algorithm 7** `ValueAc`$(s)$

---

1: **for all** children $s'$ of $s$ **do**
2:     `ValueTr`$(s')$
3: $\hat{v}_{d(s)}^{\text{comp}}(s) := \hat{r}_{d(s)}(s)$
4: **if** $m_{d(s)}(s) > 0$ but $m_{d(s)+1}(s') = 0$ for some child $s'$ of $s$ **then**
5:     $a_{d(s)}(s) := 1$
6:     $\hat{v}_{d(s)}^{\text{inc}}(s) := \hat{r}_{d(s)}(s)$
7: **for** $d := d(s) + 1, \ldots, d^*$ **do**
8:     $\hat{v}_d^{\text{comp}}(s) := \max_{s' \text{ children of } s} \hat{v}_d^{\text{comp}}(s')$
9:     $a_d(s) := \max_{s' \text{ children of } s} a_d(s')$
10:     $\hat{v}_d^{\text{inc}}(s) := \max_{s' \text{ children of } s \text{ with } a_d(s)=1} \hat{v}_d^{\text{inc}}(s')$

---