[Reviews · NeurIPS 2014]

Submitted by Assigned_Reviewer_18

This paper expands the optimistic planning literature to handle stochastic rewards. They also provide a very flexible analysis, showing how their algorithm gracefully declines in performance as the problem grows more complex.

The paper had some good ideas, but it wasn't very clearly written. Having stochastic rewards adds more technical difficulty, without adding much more depth to the problem (as usually there is enough noise in state transitions that this is not a big deal). In fact, you can always represent a discrete reward by adding transitions to more states.

I liked how the paper reused sequences of observations for different policies. What I was unclear about was whether these observations could be used across rounds of the algorithm, or whether this would make the bounds weak. I hope that they can clarify this for the final version of the paper.
Summary: This paper has some good ideas. It is somewhat incremental, and it is hard to read, but it tackles an interesting problem, and is very thorough in its analysis. It leaves little unanswered in this line of research.

Submitted by Assigned_Reviewer_38

The paper studies the planning problem with a total discounted reward objective. Given a generative model, the authors propose an algorithm and show that after a relatively small number of calls to the model, with high probability the method returns an epsilon-optimal action at the current state.

Paper [5] studies the same problem but under the stronger assumption that a full model of the rewards and transitions is known. Thus the main contribution is replacing the full model with a generative model. This is a modest improvement, but not a surprising result. In any case, the authors should discuss the challenges and why this extension is not trivial. Unfortunately no experimental studies are reported, so usefulness of these methods in real world problems is not clear.

Illustration of the algorithm is shown only in the appendix (Figure 1). As the algorithm is not straightforward, I suggest that authors move this figure to the main body of the paper.

=======================

After author rebuttal:

As this is a theory paper, please have a discussion on theoretical challenges and what makes the analysis nontrivial.
Summary: This is a modest improvement, but not a surprising result. Also, no experimental studies are reported.

Submitted by Assigned_Reviewer_43

The paper considers the problem of online planning in a MDP with discounted
rewards. More precisely, the problem is, for any given initial state, to decide
an action (at the initial state) that is epsilon-optimal with high probability.
The algorithm is assumed to have access to the generative model, i.e.,
an oracle that, when invoked with a state-action pair, provides a reward and
a next-state according to the probability distributions associated with the
target MDP. The paper designs an algorithm called StOP with sample complexity
nearly as small as the best baseline algorithm OP-MDP, which is assumed to
have access to the true distributions over rewards and next-states. Thus,
the advantage of StOP is obvious.
The sample complexity of StOP derived depends on a parameter kappa, which is
a complexity measure of the target MDP. Intuitively, kappa measures hardness
of approximating the optimal reward using random rollouts in the look-ahead
(action-transition) tree.

The paper is well-motivated and clearly written. The sample complexity bound
derived is good. It seems that the idea behind the algorithm is a reduction
to the best-arm identification, which is a standard way of estimating the game
value by random playouts in the game tree (c.f. UCT). So it should be
clarified what is the technical contribution and what is the key to deriving
the small sample complexity.

Some specific comments:
It is unclear how the authors derive from Theorem 3 the sample complexity
in terms of kappa given in Introduction. Indeed, kappa is not defined in
the body of the paper.

It would be helpful to give intuitions about the meanings of kappa and/or
S^{epsilon,*}. It is important to characterize (or upperbound) the kappa
in terms of some simpler and combinatorial measures of MDP, and thereby
to make it clear what structures of MDPs make the proposed method effective.
Summary: A well-motivated and solid work. More explanations are needed about the
technical contribution of the methods and the significance of the results.

Author Feedback
Author rebuttal: A general remark: we are very thankful to the reviewers for their work. We feel that the reviews have given us valuable feedback that we intend to use to improve the clarity of our paper.

***********************
Response to Reviewer 18

It is indeed true that any MDP with discrete random rewards can be reformulated as an MDP with deterministic rewards. Unfortunately, restricting ourselves to deterministic rewards would not allow us to simplify the algorithm and its analysis significantly (although we agree that this would be of great benefit to our paper). This is because both work with the sequence of rewards along trajectories, which would remain random, and not directly with the (then deterministic) rewards. Also, we think that it is an advantage to allow the algorithm to handle random rewards directly, as the described transformation can make the problem much more costly for StOP. As an extreme case take a bandit domain. Such a problem is relatively easy to solve using StOP as it is presented now, but the huge MDP that results from the transformation would be a lot more costly to handle.

Sharing observations is indeed at the heart of the algorithm, and can occur in multiple ways. Firstly, within a single execution of the algorithm, samples never need to be discarded. Instead, samples are steadily added to existing and newly created sample trees to satisfy the requirements of ever tighter bounds. Also within a single execution of the algorithm, samples belonging to some trajectory are reused to evaluate every policy that can realize this trajectory. The basic idea that makes this work is to lower the \delta in such a way that the union bound still holds. The exact same idea (i.e, dividing \delta by the length of the horizon, or by (r\pi)^2/6 in round r) can be used to share samples across subsequent executions of StOP, thereby making it possible to reuse all relevant samples (those belonging to the successor state encountered after taking an action) for a warm-start. We will make this more clear in the paper.

***********************
Response to Reviewer 38

In contrast to the OP-MDP [5] setting, StOP has to work with sampling. Instead of being provided with exact values for transition probabilities, it forms estimates and confidence bounds. The more often an action is sampled, the more accurate the estimates get, however, excessive sampling on branches that do not belong to good policies must be avoided in order for the algorithm to stay competitive. Finding a smart way of allocating the sampling budget, one that strikes the right balance, is not trivial.

Our main contribution is an algorithm that achieves a polynomial sample complexity in terms of \epsilon (which can be regarded as the leading parameter in this problem), and which is, in terms of this complexity, competitive to other algorithms that can exploit more specifics of their respective domains. The paper is thus focused on the theoretical problem of determining the sample complexity of planning in the general case. We agree that it is desirable to have a numerical comparison of StOP to other algorithms (and it is, indeed, our plan to do this), but this requires further engineering, and a reasonably long and deep discussion of the result. We think, therefore, that it is more adequate to perform this comparison in a follow-up paper.

***********************
Response to Reviewer 43

StOP is, indeed, based on the idea to reduce planning to best-arm identification. However, as opposed to, for example, the UCT algorithm, StOP identifies policies as arms, and deals with the fact that these arms are correlated. In this sense, it is similar in spirit to OLOP. However, OLOP is designed for the open-loop scenario, where the state transition is a deterministic function of the previous actions. The key contribution of the paper is to show how this can also be done in a closed-loop setting. The optimistic "arm" selection is more or less straightforward; the key ingredient is the sharing of all the relevant information with the correlated arms, and the evaluation of the arms base on this information.

It is true that the role of S^{epsilon,*} should have been discussed more thoroughly. To this end, caligraphic N (\calN) needs to be discussed first. It is already explained in the paper that \calN is used to formulate a high probability bound on the number of trajectories that go through some "small probability" node (as opposed to the "high probability" nodes, the Bernstein-like bounds don't work for these). \calN is, in some sense, the size of the smallest boundary that separates the high probability nodes from the low probability nodes (it doesn't actually have to be a boundary: a small part can be left uncovered). The size of the boundary is computed in terms of the expected size of the planning tree below it. This is made precise by the definition. Then, S^{epsilon,*} is simply the set of nodes "above" the boundary (i.e., the ones with high probability).

The formal definition of kappa is given in the supplementary material, appendix E, as (roughly) the limsup_{\epsilon \to 0} of (4) multiplied by \epsilon^2(1-\gamma)^2. It can be interpreted as the rate at which the "significant part" of the infinite planning tree grows. The derivation of (1) is a straightforward computation in the supplementary material, based on Theorem 3 and the definition of kappa. Also in the appendix, (along with some simplification of the definition) several upper bounds are shown for kappa in terms of K, N, and the reward structure.

We are thankful to the reviewer for pointing out how important it is to add these details to the main part of the paper. We will include these details, and the formal definition of kappa, in the final version of the paper.